# Unveiling the Immunogenicity of Ovarian Tumors as the Crucial Catalyst for Therapeutic Success

**DOI:** 10.3390/cancers15235694

**Published:** 2023-12-02

**Authors:** Galaxia M. Rodriguez, Edward Yakubovich, Barbara C. Vanderhyden

**Affiliations:** 1Cancer Therapeutics Program, Ottawa Hospital Research Institute, 501 Smyth Road, Ottawa, ON K1H 8L6, Canada; garodriguez@ohri.ca (G.M.R.); eyaku027@uottawa.ca (E.Y.); 2Department of Cellular and Molecular Medicine, University of Ottawa, 451 Smyth Road, Ottawa, ON K1H 8M5, Canada

**Keywords:** classic HLA I, non-classic HLA I, ovarian cancer, tumor immunogenicity, EMT, tumor-associated antigens

## Abstract

**Simple Summary:**

Tumor immunogenicity is one of the main factors influencing responses to cancer therapies and long-lasting antitumoral immunity. In this review, we will cover how classic and non-classic Major Histocompatibility Complex (MHC) class I molecules influence tumor composition and prognosis. Moreover, we explore other components of the tumor microenvironment such as the epithelial to mesenchymal transition of ovarian cancer cells. Then, we highlight key therapeutic strategies to overcome the lack of ovarian cancer immunogenicity and we address some open questions remaining that need further investigation in this field.

**Abstract:**

Epithelial ovarian cancer (EOC) is the most lethal gynecologic cancer. The disease is often diagnosed after wide-spread dissemination, and the standard treatment combines aggressive surgery with platinum-based chemotherapy; however, most patients experience relapse in the form of peritoneal carcinomatosis, resulting in a 5-year mortality below 45%. There is clearly a need for the development of novel treatments and cancer immunotherapies offering a different approach. Immunotherapies have demonstrated their efficacy in many types of cancers; however, only <15% of EOC patients show any evidence of response. One of the main barriers behind the poor therapeutic outcome is the reduced expression of Major Histocompatibility Complexes class I (MHC I) which occurs in approximately 60% of EOC cases. This review aims to gather and enhance our current understanding of EOC, focusing on its distinct cancer characteristics related to MHC I expression, immunogenicity, antigen presentation, epithelial-to-mesenchymal transition, and various ongoing immunotherapeutic strategies designed to stimulate antitumor immunity.

## 1. Overview of MHC Class I and Class II Molecules

Tumor immunogenicity is largely dependent on the expression of Major Histocompatibility Complexes class I and II [MHC I, II, also known as Human Leukocyte Antigen complexes I, II (HLA I, II)]. These complexes are essential proteins capable of presenting foreign antigens or self-peptides to T lymphocytes for immunosurveillance, and also for tissue homeostasis in autoimmune and infectious diseases [1]. MHC I molecules are present in almost all nucleated cells in the human body, and are differently expressed in terms of the level of transcription, transduction, and epigenetic regulation [2]. In humans, the HLA locus is found in the short arm of chromosome 6, comprising three different loci named class I, class II, and class III [3,4]. Inherited in a Mendelian fashion, the HLA gene is the most complex and polymorphic system that exists in the human genome, being associated with more than 100 different diseases, particularly autoimmune disorders [1,5]. HLA class I molecules are composed of highly polymorphic and ubiquitously expressed classical HLA-A, -B and -C allotypes, and non-classical and less polymorphic HLA-E, -F, -G, -H, -J, -K, and -L allotypes (Figure 1). Other non-classical MHC class I related molecules include: Cluster of differentiation 1 (CD1), zinc-α2-glycoprotein (ZAG), neonatal Fc receptor (FcRn), MHC class I chain-related (MIC), endothelial Protein C Receptor (EPCR) and MHC class I-related molecule 1 (MR1), which can also bind to and present small molecules such as lipids, glycolipids, metabolites and modified peptides [6]. As discussed later, non-classic HLAs have a restricted expression pattern. To this day, there are approximately 25,228 different HLA class I alleles and 10,592 HLA class II alleles that have been sequenced and named [7].

Classical and non-classical HLA I molecules form a heavy chain, presented in a glycosylated form on the cell surface, bound by non-covalent association to the invariant light chain β-2 microglobulin (β2M) which is coded in chromosome 15. The heavy chain makes three different domains (α1, -2, and -3) in the extracellular domain where α1 and α2 form a groove composed of hypervariable regions [9]. Peptides predominantly generated in the cytosol are transported to the endoplasmic reticulum through the transporter associated with antigen presentation (TAP) where other proteins such as tapasin mediate the binding of peptides within a range of 8–15 mer [10,11,12] to form an immunogenic peptide-MHC I complex (pMHCI). This complex is presented on the cell surface where it is potentially recognized by T cell receptors (TCR) on CD8+ T lymphocytes (CTLs). Some HLA I allotypes, such as a subgroup of the HLA-B locus, HLA Bw4, and HLA-Cw (HLA-C1 and HLA-C2) can engage with natural killer (NK) cells to produce an inhibitory signal [13,14,15] while HLA-E and -G allotypes can interact directly with CD94/NKG2 receptors on NK cells, inhibiting or inducing their activation [16,17,18] (Figure 1).

Similarly, HLA II complexes possess two polymorphic chains composed of five isotypes designated as HLA-DM, -DO, -DP, -DQ, -DR, with a more restrained expression. Peptides varying from 13–25 mer [19] derived from an extracellular origin can bind the MHC II groove to form a pMHCII complex in antigen presenting cells (APCs) including B lymphocytes, dendritic cells (DCs), macrophages, monocytes, Langerhans cells, endothelial cells, thymic epithelial cells, activated T lymphocytes, and some epithelial cells found in the cervical and colorectal regions, which can be recognized by TCR on CD4+ T lymphocytes [20,21,22]. Classical HLA class I, and to a greater extent HLA class II, can be detected in a soluble form (sHLA) in plasma, urine, and various other bodily fluids in healthy individuals [23,24].

The class III HLA region comprises more than 50 genes encoding for proteins not only involved in immunity (activation of complement, inflammation, immunoglobulin superfamily members, and cell stress) but also in hormonal synthesis, and extracellular matrix organization [25,26,27], the discussion of which is beyond the scope of this review article.

## 2. Ovarian Cancer Immunogenicity

As in most solid tumors, EOC cells downregulate MHC I expression as an immune evasion mechanism. Indeed, expression of MHC I genes is impaired in up to 60% of ovarian tumors [2,28,29]. Several EOC subtypes including serous, clear cell, endometrioid and mucinous, are immunogenic tumors capable of recruiting T cells into the tumor microenvironment (TME), resulting in positive prognoses [30,31,32,33]. Indeed, the presence of T cells specific to neoantigens expressed by EOC cells is strongly associated with increased survival [34,35] and the mechanisms related to immune cell infiltration are dependent on the antigen processing and presentation machinery (APM) components and MHC-I and -II status [36,37]. Nevertheless, the heterogeneity of HLA I allotype expression in a healthy cell is ultimately lost as tumors evolve to express fewer allotypes or completely lose HLA I expression [2,32,36].

The mechanisms by which MHC I expression is suppressed during tumor development has a major impact on the response to cancer immunotherapy [38]. Cancer cells lose or downregulate MHC I molecules because of the loss or decreased transcription of MHC I related genes or defects in APM components [39]. These defects can be classified as either “hard” or “soft”, depending on whether they are irreversible or reversible, respectively, by gene regulators or cytokines [39]. In healthy cells and cancer cells with soft defects, APM and MHC I genes can be induced by the IFN regulatory factor 1 (IRF-1), NF-κB, and the NOD-like receptor family caspase recruitment domain-containing 5 (NLRC5) in response to stimulatory cytokines such as TNF-α and IFN-γ [40,41,42].

EOC immunogenicity has been measured with humoral and cellular antitumor immune response markers detectable in peripheral blood, tumor sites, and ascites derived from EOC patients [43]. Goodell and colleagues were able to detect p53 antibodies in serum from 104 EOC patients, whose levels were positively correlated with overall survival [43]. Importantly, the presence of neoantigen-reactive T cells in patients with EOC can improve survival [34,35]. Brown et al. analyzed TCGA RNA-seq data from six EOC tumor sites in 515 patients, and identified mutational epitopes presented by the autologous HLA-A alleles that predicted tumor immunogenicity. These mutational epitopes triggered higher CTL content in the tumor niche and were associated with increased patient survival. However, tumors devoid of CTL infiltration lacked these mutational epitope signatures [34]. Wick and colleagues analyzed T cell reactivity towards 79 Tumor Associated Antigens (TAAs) originating from non-synonymous mutations identified by whole exome sequencing of autologous tumors, using T cells from the tumors of three EOC patients. A robust and specific CD8+ T-cell response to the mutated hydroxysteroid dehydrogenase-like protein 1 (HSDL1)^L25V^ was detected in one patient at different levels over the course of disease recurrence, highlighting the evolving expression of neoantigens and the limit of naturally occurring antitumoral immunity recognition over EOC progression [35].

In fact, ovarian tumors generally possess intermediate or low mutational burdens as a consequence of a very low incidence of naturally processed and presented neoantigens that could generate a significant antitumoral response [44]. Nonetheless, TAA presentation is the pivotal factor enabling CTL-tumor cell recognition and killing [45]. The following section will elucidate the current understanding of the expression of HLA class I allotypes and their intricate association with tumor burden and survival outcomes.

### 2.1. Classic HLA Class I

Downregulation of classic MHC I is a prevalent immune evasion mechanism used by tumor cells to escape antitumor T-cell-mediated immune responses [46]. Under physiological conditions, classic HLA class I molecules are expressed by virtually all cell types, allowing for NK or T cell recognition to achieve immunosurveillance. A tissue microarray of 339 EOC samples stained for MHC I and β2M established a positive correlation between HLA I expression and increased patient survival independent of age, stage, level of cytoreduction, and exposure to chemotherapy [47]. Although specific allotypes, such as the HLA-A*02 subtype, correlate with poor prognosis in advanced-stage serous EOC [8], the HLA-B allotype is a positive predictor of the immune response to cancer testis’ TAAs [48]. While MHC I gene expression in EOC cells can be downregulated as a consequence of somatic mutations, these mutations are not common in EOC. Shukla et al. analyzed 7930 samples across 20 different tumor types and found that ovarian carcinoma, glioblastoma, and breast cancer largely lacked somatic mutations in HLA genes, being present in only 0–0.6% of the tumor samples [49].

Despite the lack of HLA mutations, differences related to total HLA I and II expression in ovarian tumors occur. Using RNA-seq, immunohistochemistry, and flow cytometry analysis on 27 EOC samples, Schuster et al. revealed that most ovarian tumors display strong HLA I expression, and to some extent HLA II expression. However, only the EpCAM+ population was considered in the cancer cell subset which may not necessarily represent most EOC cells [50], and the degree of immune infiltration of the EOC samples was not included in the analysis, potentially resulting in an overestimated HLA expression in highly infiltrated tumors. In a more recent study, tissue sections from 30 untreated high-grade serous ovarian cancers (HGSC) were analyzed for MHC I staining and showed sub-clonal loss in 7/30 (23%), including areas of retained MHC class I expression immediately juxtaposed with areas of negative staining [51]. Neither of these studies classified the overall diversity of HLA class I allotypes being retained in the tumor tissue, which may turn out to be a notable weakness, as non-classic HLA class I expression may negatively impact patient survival as described in the following section.

### 2.2. Non-Classic HLA Class I

As cancer cells downregulate classic HLA I molecules to avoid CTL recognition, they can also circumvent detection and elimination by NK cells through alternative means. Non-classic HLA I molecules are less polymorphic and display distinct expression patterns in developing and adult tissues, exerting functions in both the innate and adaptive immune systems [52,53]. In many malignancies, non-classic HLA I allotypes are aberrantly expressed, perhaps as a consequence of the proximity of genes such as HLA-E, -F, and -G to the class I region on chromosome 6 [54]. Indeed, aberrant expression of non-classical HLA I molecules in tumors contributes to inhibition of NK cells, rendering tumor cells resistant to NK cell-mediated lysis [55]. The following sections summarize key studies underlining the potential effects of non-classic HLA molecules in EOC.

#### 2.2.1. HLA-E

HLA-E is expressed in most healthy human tissues, including placenta, but with a weak expression pattern on the cell surface [56]. HLA-E/peptide complexes are recognized by the CD94 receptor in conjunction with the inhibitory NKG2A or the stimulatory NKG2C molecule, expressed on the majority of NK cells and some activated CTLs [16,57,58]. In several malignancies, HLA-E can compensate for the loss of classic HLA I expression. Indeed, HLA-E expression is upregulated concurrently with the downregulation of classic HLA I allotypes and the presence of free β2M in the cytoplasm of tumor cells [55]. Tumor cells possessing an imbalance in heavy chain and β2M expression also possess this unique inverse expression pattern. HLA-E/β2M complexes are weaker compared to classic HLA I complexes, and when the latter are absent the HLA-E complexes become prevalent [55].

Nonetheless, in other cases, HLA-E overexpression can unbalance pre-established antitumoral immunity. In a study including 150 cervical and 270 EOC samples, HLA-E was found to be expressed at higher levels than healthy tissue and positively associated with expression of APM components, classical HLA I molecules, and CTLs in 80% of the samples [59]. In situ analysis revealed that HLA-E interacts with the inhibitory CD94/NKG2A receptor predominantly expressed on intraepithelial CTLs. Notably, the favorable prognostic effect of infiltrating CTLs in EOC was neutralized by high expression of HLA-E on the surface of the cancer cells, suggesting that HLA-E impedes antitumoral CTL activity in the TME [59].

Interestingly, a recent multivariate analysis from the phase III AGO-OVAR-12 trial involving 103 HGSC patients suggested that the HLA-E/CD94-NKG2A/2C axis is a potential target to improve antitumoral activity, particularly in the group of patients with homologous recombination deficiency (HRD). Similar to the overexpression of HLA-E in unstable microsatellite tumors in colorectal cancer [60], HLA-E was preferentially overexpressed in HRD HGSC, although the germline or somatic BRCA mutation status was not explored in this study [61]. HGSC patients with a high fraction of intratumoral CD3+ T lymphocytes had longer progression-free survival (PFS) as well as high HLA-E expression on tumor cells, along with an HRD profile which showed improved overall survival [61]. Moreover, the authors found that HLA-E-overexpressing tumors were highly enriched in Tregs (FOXP3+, ICOS+) and IgG, and, similar to findings by Gooden and colleagues [59], the survival benefit driven by T cell infiltration was lost with high HLA-E expression. This study provides insights into the impact of HRD lesions enhancing genomic instability which also influence tumor immunogenicity, tumor immune infiltration, and, potentially, HLA expression. Although, it is still not clear if genomic instability can directly affect HLA-E expression as a consequence of DNA damage [62], or if these findings are a consequence of type II IFN (IFN-γ) production in the TME, as HLA-E can be induced by IFN-γ [63].

Genetic variations in HLA-E alleles can influence their role in tumor immunosurveillance. Only two alleles (HLA-E*0101 and HLA-E*0103) have been reported, and they potentially accomplish different functions [64]. Zheng et al. studied 85 primary serous EOC tumors compared to 100 healthy tissues and found a high frequency of HLA-E*0103 expression at the transcriptional and protein levels in serous EOC. This allele improved the transfer of the HLA-E molecule to the cell surface, rendering the HLA-E/peptides complex more stable and increasing its capability to inhibit NK cell cytolysis [64]. However, it is still unclear how the HLA-E*0103 allele is preferentially expressed in EOC tumor cells and how both alleles are affected and regulated during tumor development.

#### 2.2.2. HLA-F

HLA-F is the smallest of the HLA I molecules and is expressed in the skin, the developing fetal liver, to a lesser extent in the placenta and extra-placental tissues, and also in monocytes and lymphocytes such as NK, T, and B cells [65]. HLA-F is expressed as an empty heterodimer devoid of peptide in the cytosol, and acts as a ligand for several intracellular proteins such as TAP and calreticulin [66], and immune specialized receptors including immunoglobulin (Ig)-like transcript 2 (ILT2), ILT4 [67], KIR three Ig domains and long cytoplasmic tail 2 (KIR3DL2), KIR two Ig domains and short cytoplasmic tail 4 (KIR2DS4) [68], and KIR three Ig domains and short cytoplasmic tail 1 (KIR3DS1) [69]. Interestingly, HLA-F can also participate as a chaperone in the cytosol to stabilize classic HLA I open conformers (without peptide) on activated monocytes and lymphocytes, cooperating in the exogenous cross-presentation pathway independent of the TAP and tapasin proteins [70]. Collectively, the evidence suggests that HLA-F is an immune regulatory molecule that acts as a stabilizer; however, its binding partners are unknown.

HLA-F mRNA is overexpressed in glioblastoma compared to healthy tissue [71]; however, to date, there is no known link between HLA-F and EOC. Nevertheless, Fang et al. recently found that the long noncoding RNA HLA-F-AS1 is overexpressed in EOC cells and attenuates EOC development in vivo and in vitro by targeting the miR-21-3p/PEG3 axis [72]. Since HLA-F plays several immune regulatory roles, more studies are needed to better understand its potential role in EOC tumorigenicity.

#### 2.2.3. HLA-G

Similar to HLA-E, HLA-G is expressed in extra-embryonic tissues during gestation, especially in the placental trophoblasts where it participates in the establishment of an immunotolerant state during pregnancy [56,73]. HLA-G can also be found in the cornea, nail matrix, pancreas, erythroid and endothelial precursors, and stem cells [74,75,76]. In the thymus, HLA-G appears to participate in the development of the T cell repertoire, potentially explaining T cell tolerance to HLA-G [77]. In healthy tissues, HLA-G plays a protective immunosuppressive role, whereas, under neoplastic conditions, HLA-G allows tumor progression by being overexpressed in cancer cells [78]. This allotype can undergo alternative splicing of its primary transcript to produce seven HLA-G protein isoforms (HLA-G1 to -G7), three of which can become soluble proteins (HLA-G5 to -G7) [79]. HLA-G possesses a heterogeneous and focal expression pattern, and can be expressed at the cell surface, secreted, or associated with tumor-derived exosomes. In all these forms, HLA-G exerts immune-modulatory functions by binding to CD8, LILRB1 (Leukocyte Immunoglobulin-Like Receptor B1, expressed by monocytes, DCs, B cells, and NK cells), LILRB2 (expressed only by monocytes), and KIR2DL4 (expressed by placental NK cells) [79]. HLA-G acts as a tolerogenic molecule by inhibiting the immune cell functions of APCs, NK cells, and CD4+ and CD8+ T cells by directly binding with their inhibitory receptors or indirectly through trogocytosis, leading to T cell anergy and rendering T lymphocytes more regulatory and immunosuppressive [8]. sHLA-G also induces apoptosis in NK cells and antigen specific CD8+ T lymphocytes [80].

In view of the numerous immune-regulatory functions of HLA-G, it is not surprising to find that its expression is associated with a worse clinical outcome in patients with solid tumors, including mesothelioma and breast carcinoma [81,82], but not in hematological malignancies [78]. Interestingly, in vitro studies have shown that just 10% of HLA-G+ tumor cells is sufficient to protect the rest of the tumor cells from elimination by CTLs [83], highlighting the strong regulatory capabilities of HLA-G in tumor promotion. When Lin et al. transfected the EOC cell lines HO-8910 and OVCAR-3 with the HLA-G gene, the cells acquired higher invasion potentials compared to parental cells. Moreover, when introduced into Balb/c nu/nu mice, EOC cells overexpressing HLA-G developed widespread metastasis, conferring poor survival [84]. HLA-G is also involved in tuning the immune response, as when HLA-G+ cells were cultured with PBMCs, the immune response was more accentuated towards a Th-2 cytokine profile [85]. This is the opposite of the actions of sHLA-G, which favors an anti-inflammatory environment induced by the release of IL-10 [86].

HLA-E and HLA-G allotypes can be co-expressed in EOC tissues, with a semi-quantitative analysis of 62 EOC revealing high HLA-E expression associated with the serous subtype and advanced stages [87]. In another study, Andersson et al. analyzed non-classic HLA I expression in primary tumors from 72 patients with advanced-stage serous EOC and in metastatic cells derived from ascites from eight patients [8]. The site-specific downregulation of classical MHC I allotypes alongside the focal cell expression of HLA-G and HLA-E correlated with poor survival and worse prognosis in patients harboring the HLA-A*02 subtype, but not with different HLA genotypes. Interestingly, metastatic lesions had a higher expression of HLA-G compared to primary tumors, which was inversely correlated with the frequency of TILs and increased immunosuppression [8]. Furthermore, sHLA-G may be a potential marker of malignant ascites in EOC [88] which could be used to assess the progression and recurrence of the disease [89]. HLA-G may also regulate vascular remodeling in tumors, pointing towards the strong capability of this molecule to influence the EOC TME [90].

In contrast, Rutten et al. showed that HLA-G expression was correlated with longer PFS and overall survival and an improved response to chemotherapy in 169 HGSC patients [91]. Importantly, serum sHLA-G levels did not correlate with protein or gene expression levels in the tumors or survival [91]. Discrepancies regarding the pro-tumoral or anti-tumoral effects of HLA-G remain controversial and yet to be clarified. According to the available studies, HLA-G is frequently expressed in high-grade ovarian tumors, especially at advanced stages, and in rare cases in low-grade tumors [92]. Contradictory findings can be due to differences in staining techniques, gene expression vs. protein expression, scoring scale, or the definition of positive expression in each study. In addition, the role of HLA-G could be a consequence of the heterogeneity, unique to each EOC, being influenced by the tumor mutational burden (TMB) and the immune composition, in some scenarios, compensating for a lack of classic HLA I expression, and in other scenarios being aberrantly co-expressed with HLA-E. Moreover, since there are several spliced forms of HLA-G, post-translational regulation within the TME may play a dominant role in protein expression which could ultimately change the impact of HLA-G expression (membrane-bound or soluble) and function in the tumor niche.

### 2.3. NLRC5, the Master Regulator of MHC Class I Expression

NLRC5 (also known as CITA) is a critical regulator of MHC I genes, as well as some related genes involved in MHC I-dependent APM via the formation of CITA enhanceosomes [41,93,94,95]. NLRC5 induces the expression of both classical and non-classical class I molecules, but also the main components of the APM pathway like β2M, immunoproteasome components (PSMB9, i.e., LMP2), and TAP1 [41]. The expression of MHC I and APM components strongly correlates with NLRC5 gene expression in multiple cancers such as lung, melanoma, thyroid, breast, prostate, uterine, and EOC. Defects in NLRC5 expression found in human tumors include genetic modifications such as copy number loss, somatic mutations, and promoter methylation, which strong downregulate MHC I expression [96,97]. An analysis of the NLRC5 gene in multiple cancer types revealed that EOC patients (*n* = 489) displayed the highest frequency of copy number loss at 72.2%. This loss was associated with the reduced expression of NLRC5 and MHC I and related genes, including HLA-A, HLA-B, HLA-C, B2M, LMP2, and LMP7 [96].

In summary, classic HLA I expression is associated with better survival for EOC patients while non-classic HLA I is more pronounced in aggressive and more advanced EOCs. Many unknown aspects related to non-classic HLA I, such as ligands, polymorphisms, and post-translational regulation, still need to be explored. Understanding these factors will help us to better grasp their influence on the EOC TME and, in particular, how they influence the response to treatment.

## 3. Other Tumor Microenvironment Factors Influencing Ovarian Cancer Immunogenicity

The EOC TME is highly immunosuppressive, frequently containing a tumor promoting network of cytokines, such as IL-10 and TGF-β, and also pro-inflammatory factors such as TNF-α [98]. In the following sections, these factors will be discussed in the context of the epithelial–mesenchymal transition (EMT) as increasing evidence suggests that these cytokines can influence HLA class I expression and overall tumor immunogenicity.

### 3.1. EMT Effects on HLA Expression in Cancer

Downregulation of HLA I expression has been linked to the EMT in melanoma, colorectal carcinoma, prostate adenocarcinoma, breast carcinoma, and EOC [99,100]. The EMT is a process through which epithelial cells shed cell–cell junctions, detach from the basement membrane, and undergo transcriptional and morphological changes to acquire mesenchymal characteristics and gain stem cell-like features such as the capacity for self-renewal [101,102]. Enhanced capacity for migration and invasion enables the cells to extravasate into the bloodstream and form metastatic colonies in peripheral tissues [103,104]. In prostate cancer cells, overexpression of the EMT transcriptional regulator Snail, or treatment with EMT-inducing Transforming Growth Factor Beta 1 (TGF β1), reduced the expression of HLA class I molecules [105]. Similar findings have been found in breast [106] and pulmonary cancers [107]. EpCAM+ EOC cells show high levels of the expression of MHC I classical haplotypes when compared to benign fallopian tube samples [108]. EpCAM is silenced in mesenchymal cancer cells [109]; this suggests that cancers that retain their epithelial features can also retain high levels of expression of the classical MHC I haplotypes.

In syngeneic tumors from mesenchymal cell lines established in the MMTV-PyMT mouse model of breast carcinoma, there was a remarkable decrease in MHC I expression alongside an increase in the checkpoint inhibitor PD-L1 compared to tumors arising from epithelial MMTV-PyMT cell lines expressing EpCAM and E-cadherin [106]. Evidence from this study and others points to the association of EMT with downregulation of MHC I components, whereby mesenchymal cells are able to avoid detection by immune cells by downregulating HLA and APM.

Strategies to bypass the negative effects of EMT on MHC I machinery have been proposed [110]. One possible strategy is to target EZH2, an epigenetic regulator whose activity is correlated with EMT in breast cancer and melanoma [111,112,113]. The combination of the EZH2 inhibitor GSK126 with anti-PD-1 in the anti-PD-1 resistant MOC1-esc1 mouse tumor model of head and neck squamous cell carcinoma reduced the growth of those tumors while also causing an increase in MHC I expression. This suggests that EZH2 expression synergizes with the immune checkpoint PD-L1 to reduce APM [114]. GSK126 treatment has also been found to upregulate classic MHC I allotypes in a human head and neck cancer cell line, suggesting similar effects to EZH2 in human cancers. Similarly, inhibition of the EMT in the highly metastatic 4T1 tumor model using an angiokinase inhibitor was found to upregulate MHC I expression and APM machinery [115]. It appears that the inverse relationship can also hold true, as the upregulation of the MHC I machinery protein B2M induced EMT in B2M-overexpressing clones of MCF7 (breast), H358 (lung), and SN12C (renal) cancer cells both in vivo and in vitro [116]. This suggests a complex relationship of bidirectional influences between MHC I, APM, and EMT in cancer cells, with EMT regulators such as EZH2 having a dampening impact on the machinery.

Despite the strong evidence for an association between EMT and the loss of MHC I/APM expression, this link is not universal. In a mesenchymal breast cancer cell line, a mesenchymal-to-epithelial reversion induced by upregulation of an EMT suppressor microRNA miR-200 [117] also suppressed PD-L1 expression and led to greater CD8+ T cell cytotoxicity. However, in a bioinformatic analysis of microRNAs affecting APM expression, high levels of the expression of miR-200a-5p suppressed TAP1 expression in melanoma [118]. Another analysis revealed that transgenic knock-in of miR-200c increased MHC I expression in murine mammary cancer cell line EO771 [119]. Thus, while the roles of these individual miR-200 molecules in suppressing EMT are well-established [120], they appear to have differing roles in the regulation of HLA and APM expression, which warrants further investigation.

Remarkably, a pan-cancer gene signature for epithelial–mesenchymal plasticity generated in our lab [121] includes genes for HLA-A, -C, and -E, TAP1, PSMB9, and B2M, suggesting that MHC I machinery is, at least in part, associated with the EMT in a variety of cancer cell lines and tumors from different origins. We leveraged this compiled single-cell dataset to compare HLA I gene expression against the average expression of a set of EMT genes, using either the cancer cell specific gene module [121] or the classical EMT hallmark module [122]. The results show a positive correlation for both classic and non-classic HLA I molecules (HLA-A-C/E-G) in eight different cancer types (Figure 2).

For EOC, this correlation holds true for most of the individual HLA I allotypes (Appendix A). It is possible that this altered association between EMT and HLA expression is due to the fact that these are data from human tumors rather than from cell lines in vitro or mouse models, which suggests that the TME may well influence this relationship. Taken together, these findings suggest that the presence of HLA I molecules in cancer cells is associated with the presence of mesenchymal cells, and the EMT may promote the expression of HLA I molecules in certain cancer disease.

### 3.2. EMT Inducers and HLA I Expression

#### 3.2.1. TGF β1

TGF β1 is an established regulatory cytokine responsible for the induction of EMT in cancer cells [123] that can maintain mesenchymal states in an autocrine fashion [124,125]. After binding to TGF β receptors (TGFβR), SMAD-signaling takes over and R-SMADs and co-SMAD complexes translocate to the nucleus to induce transcription of EMT transcription factors (EMT-TFs) among other target genes [126]. TGF β1 has been previously linked to an attenuated expression of HLA I molecules and APM components in mouse CCK168 cells in vitro or in epithelial PyMT cells in vitro [106,127,128]. Importantly, TGF β1 targets the APM molecules in cancer cells by downregulating the H2 complex (MHC I in mouse), B2m, Tap1, and Tap2 shown in an in vitro model of CCK168 squamous cell cancer cells [127]. In human bone marrow-derived stem cells, two APM components, B2M and ERAP1, were downregulated following TGF β1 treatment [129]. Mouse PyMT tumor cells that were forced to undergo EMT through the upregulation of EMT-TFs ZEB1 or SNAIL in vitro also showed a marked decrease in B2M protein, a crucial co-factor for MHC I stability and antigen presentation on the cell surface [106]. Bearing in mind that these results were performed in murine models, the behavior of EMT and MHC I machinery may be different in human biology.

Published datasets from our lab also demonstrated a correlation between TGF β’s induction of EMT and the upregulation of EMT-TFs such as ZEB1, SNAI1, and SNAI2 in OVCA420 (ovarian), A549 (lung), and MCF7 (breast) cells [130]. In that study, we induced EMT in the cancer cell lines via treatment with TGF β1, EGF, or TNF-α for up to 7 days with analysis conducted using single-cell RNA-sequencing. We re-analyzed those data for the expression levels of MHC I components, expecting to find a reduction in APM associated with EMT, but we found few changes in the expression of HLA and AMP genes (Figure 3). It is notable that the ovarian cancer cells had very low expression levels to start, with the exception of relatively high levels of the expression of HLA-C. The expressions of HLA-B and HLA-E were modestly reduced and increased, respectively, by TGF- β1, suggesting less classical MHC I and more non-classical MHC I as the cells become more mesenchymal. While these findings may be linked to specific epigenetic and genetic features unique to each cell line, it may also reflect an EMT-induced reversion of the cancer cells to a more stem-like state, such that they upregulated the MHC-I machinery similar to pluripotent stem cells [131,132].

While the consequences of this type of shift from classic to non-classical HLA I expression has been described in detail above, it is important to consider the effect of TGF-β on non-classic HLA I expression in EOC tumorigenicity. Expression of the HLA-E receptors CD94/NKG2A can be induced in response to cytokines such as TGF-β in CD8+ T cells [133,134]. Therefore, in the EOC TME, concomitant stimulation of the TCR and exposure to TGF-β may indirectly dampen antitumoral responses due to the expression of inhibitory CD94/NKG2A receptors which can interact with HLA-E+ EOC cells. Hence, an increased expression of HLA-E in EOC cells is a mechanism used to evade immunosurveillance, not only through inhibiting NK cells but also CTLs, even in the absence of classic MHC I molecules [135].

Inhibition of Smad3 via a selective inhibitor of Smads (SIS3-HCI) in human bone marrow-derived stem cells resulted in the attenuation of TGF β1′s ability to downregulate MHC I expression, indicating the importance of SMAD2/3 signaling in mediating TGF β1’s actions on suppressing APM [129]. Whether the downregulation of APM gene expression is dependent on EMT remains unclear, and it may well be that TGF β pathway activation promotes EMT and suppresses APM via independent mechanisms. In that context, it is important to note that TGF β is usually abundant in the TME. We could find no evidence that TGF β might suppress the expression of NLRC5, a critical regulator of MHC I. However, the reverse seems possible, as a deficiency in NLRC5 in endothelial and cardiac cells from a mouse model of diabetes [136], or in the human hepatic stellate cell line LX-2 [137], showed abrogation of phosphorylated Smad2/3, suggesting a link between a loss of NLRC5 and reduced TGF β pathway signaling.

In addition to TGF-ß1-mediated suppression of MHC I, TGF β1 can also downregulate MHC II molecules through the inhibition of the class II transactivator (CIITA), a master transcription factor for MHC II genes [138,139]. Further studies are needed to elucidate the mechanisms regulating MHC I and MHC II expression in cancer cells, with or without the influence of factors associated with EMT in those cells.

#### 3.2.2. EGF

Very little is known about the effects of direct EGF stimulation on cancer cells in the regulation of MHC I/-II. However, the activation of the EGF receptor (EGFR) by a wide variety of ligands such as TGF-α, HB-EGF, amphiregulin, epiregulin, betacellulin, epigen, and neuregulin [140] has well-established effects on reducing MHC I and MHC II expression [141,142]. The signaling mechanisms involved include RAS/MAPK [142], MEK/STAT [143], and HER2 [144], with the majority of studies contextualized in breast cancer, melanoma, and mesothelioma.

In a non-cancer context, THP-1 cells infected with Brucella abortus bacteria showed a marked reduction in the expression of MHC I molecules during Western blot and flow cytometry analysis. However, blockage of the EGFR with Cetuximab rescued the expression of MHC I in these cells, highlighting the ability of the EGFR pathway and to suppress MHC I expression [145]. Skin biopsies from cancer patients receiving EGFR inhibition therapy show an increase in MHC I protein expression, as well as an increase in MHC I and MHC II allotypes at the RNA level [146]. Both of these results suggest that MHC I expression levels are reduced when the EGFR pathway is active, and this may be a consequence of the activation of EGFR by EGF or any other ligand that binds the receptor [147].

Surprisingly, we found that EGF treatment had no discernible effect on the expression of classical and non-classical HLA I allotypes in OVCA420, MCF7, and A549 cells (Figure 3). While there is ample literature linking EGFR stimulation to MHC I and MHC II expression, these results suggest that EGF itself may not be the key ligand in this process. It is also important to note that the treatment with EGF induced EMT in these cells, indicating that the process of EMT itself is not sufficient to suppress MHC I expression. Since EGF can bind to receptors other than EGFR, such as ErbB2 (HER2), ErbB3 (HER3), and ErbB4 (HER4) across most cancer contexts, and is a key player in cancer metastasis [148], it is likely that the EGF-associated mechanisms driving EMT are distinct from those regulating MHC I expression.

#### 3.2.3. TNF-α

The mechanisms underlying TNF-α-driven changes to APM are well-established [149]. In contrast to the effects of TGF β1 as a negative modulator of MHC I expression [150], TNF-α activates the NF-κB pathway, which stimulates MHC I expression in cancer [151,152]. Since TNF-α also promotes EMT, this suggests that the activation of the NF-kB pathway by TNF-α might be linked with the EMT. This suggestion is strengthened by the data shown in Figure 3C, where the upregulation of MHC I components due to TNF-α treatment of breast and lung cancer cell lines is seen to coincide with EMT across treatment timepoints. Abrogation of NF-κB activity in the human EOC cell line JHOC-5 with the NF-κB inhibitor DHMEQ restored the antitumoral activity of T cells when implanted in a mouse [153] suggesting that activation of the NF-κB pathway, perhaps through its upregulation of MHC I, may boost anti-tumoral immunity in EOC. In another case, the treatment of OVCA420, MCF7, and A549 cells with TNF-α showed an increase in the expression of HLA-A, HLA-C, TAP1, and PSMB9 in our experiments (Figure 3), suggesting that TNF-α could activate pathways, such as NF-κB, that lead to the upregulation of APM.

## 4. Therapeutic Strategies to Overcome the Lack of Immune Recognition in Ovarian Cancer

Over the past 25 years, there has been a tremendous effort to improve the response of cancers to immunotherapies to achieve tumor elimination. As the activity of CTLs and NK cells allows for successful antitumoral immunity, several approaches have been proposed to overcome immune evasion mechanisms established by the EOC TME. The following sections summarize the immune platforms under development that will potentially achieve better therapeutic outcomes for EOC. Figure 4 summarizes the therapeutic interventions aiming to increase EOC immunogenicity.

### 4.1. Targeting TAAs and HLAs

#### 4.1.1. Chimeric Antigen Receptors

CARs have been exploited to direct tumor recognition and eradication. CAR-T cell therapy has achieved successful efficacy against hematological malignancies [154]; however, there are still key limitations to address before antitumor immunity can be achieved in solid tumors. Among these limitations is the lack of TAA generation/recognition in the immunosuppressive TME [155], especially in EOC where a low mutation burden limits tumor-specific antigen generation [44]. Recent advances in targeting specific TAAs have progressed into clinical trials, with seven trials using CAR technology for the treatment of EOC currently recruiting patients [156]. Most of these studies target known TAAs that are overexpressed in EOC, including human epidermal growth factor receptor 2 [National Clinical Trial number (NCT) NCT04511871), mesothelin (NCT04562298, NCT02580747, NCT01583686), MUC16/CA125, CD276 (NCT05211557), alpha-folate receptor, Claudine 18.2 (NCT05472857) and Tumor-associated glycoprotein 72 (NCT05225363). Several TAAs are promising targets for CAR therapies and were recently reviewed by Zhang et al. [157]. Appendix A summarizes the current clinical trials addressing different components of EOC immunogenicity.

##### Human Epidermal Growth Factor Receptor 2 (HER2)

HER2 is an oncogene that promotes transformation and tumor development and is overexpressed/amplified in several cancer types [158], including EOC, with reported incidence of expression ranging from 1.8% to 76% of tumors, especially in the HGSC subtype [159]. In pre-clinical studies, Sun and colleagues showed that HER2-CAR-T-cells were able to recognize and kill HER2+ ovarian cancer cells [160], advancing this therapeutic approach for the treatment of HER2-expressing EOC into the clinical trial stage.

##### Mesothelin

Mesothelin is another TAA overexpressed in about 30% of EOC cases. Mesothelin comprises a group of glycoproteins bound to the plasma membrane which are normally found in tissues including the pleura, peritoneum, pericardium, and mesothelial cells [161,162]. Studies have already demonstrated the feasibility, safety, and clinical evidence of antitumoral responses in patients with progressive mesothelin-expressing solid malignancies treated with mesothelin-specific mRNA CAR T cells [163]. However, the overall clinical benefit in EOC patients remains yet to be proven.

##### MUC16/CA125

MUC16, also known as cancer antigen 125 (CA125), consists of a repetitive epitope of mucin glycoprotein and has been implicated in cell proliferation and tumor promotion [164]. CA125 is a serum biomarker used to monitor EOC tumor burden and recurrence [165] as it is often overexpressed, and it can be found in >80% of serous EOC cases [164]. CAR-T cells have also been generated to target MUC16 TAA, showing the specific killing of MUC16+ ovarian cancer cells in vitro and delayed tumor development or fully eradicated disease in vivo [166].

##### Alpha-Folate Receptor (FRα)

FRα is overexpressed in approximately 90% of ovarian carcinomas, and is also detected in several other malignancies [167]. Kalli et al. found FRα expression in 134 of 186 (72%) primary and 22 of 27 (81.5%) recurrent EOCs, especially in the more common high-grade subtype where its expression was primarily found in metastatic lesions [168]. FRα is an attractive TAA since about 70% of EOC and breast cancer patients showed measurable immune responses against FRα [169], and, due to its highly restricted distribution and overexpression in EOC, FRα is a suitable candidate for cancer immunotherapy. Kershaw et al. reported the first clinical trial for the treatment of metastatic EOC using genetically engineered T cells that are reactive to FRα by modifying autologous T cells with a chimeric gene incorporating an anti-FR single-chain antibody linked to the signaling domain of the Fc receptor gamma chain. Unfortunately, none of the 14 patients enrolled in this study had any reduction in tumor burden because the engineered T cells were unable to persist over time or infiltrate the tumor [170]. Ebel et al. subsequently introduced Farletuzumab (MORAb-003), a humanized high-affinity monoclonal antibody against FRα [171] which demonstrated promising preclinical antitumor activity in xenograft models. Since then, Farletuzumab has been combined with other therapies such as carboplatin, paclitaxel, doxorubicin, and taxane in several clinical trials with poor outcomes so far [NCT01004380, NCT00738699, NCT02289950, NCT00318370, NCT01018563, NCT00849667 were all halted due to the lack of efficacy based on PFS in participants]. Current clinical trials are evaluating a derivative compound, MORAb-202, which works as an antibody-drug conjugate with farletuzumab and is linked to eribulin as the cytotoxic payload with promising antitumor activity [172] (NCT05613088, NCT04300556). Another agent, Mirvetuximab soravtansine, is an antibody–drug conjugate which targets FRα and has been tested against platinum resistant EOC, where it demonstrated clinically meaningful antitumor activity and favorable tolerability and safety in patients with FRα expression [173,174,175,176].

##### Tumor-Associated Glycoprotein 72 (TAG72)

The main limitation of CAR therapies is the expression and presentation of the cognate antigen, which have been shown to evolve over time as a consequence of immunoediting, highlighting the dynamic nature of the antitumoral responses. The use of CAR-T cells with dual antigen specificity can mitigate this problem. Shu and colleagues engineered CAR-T cells targeting TAG72 and CD47 for the treatment of EOC [177]. TAG72 is an aberrantly glycosylated cell surface glycoprotein overexpressed in several adenocarcinomas [178], including EOC, where it was detected in almost 90% of the cases [179], especially in more advanced stage tumors [180]. CD47 is a “don’t eat me” signal inhibiting natural phagocytosis by macrophages, but it is also ubiquitously expressed by ovarian cancer cells. Therefore, by using a blocking antibody, CD47 facilitates elimination of cancer cells by restoring the engagement of macrophages [181]. Shu et al. demonstrated that targeting multiple antigens may improve the effectiveness of CAR immunotherapy for EOC [177]. The safety and tolerability of TAG72-CAR-T cells is currently being evaluated.

#### 4.1.2. Targeting MHC I Molecules

##### HLA-E

Humanized monoclonal antibodies targeting the HLA-E receptor NKG2A (Monalizumab) are readily available [182] and are currently being used in clinical trials in combination with durvalumab (blocking antibody impeding PD-1 binding with PD-L1) for solid tumors including EOC (NCT02671435; [183]). This represents a promising strategy for future clinical studies in HLA-E+ ovarian tumors.

##### HLA-G

HLA-G: Jan et al. developed CAR-NK cells with enhanced cytolytic effects via DAP12-based intracellular signal amplification, and a single-chain variable fragment (scFv) against HLA-G. Using in vitro and in vivo models, they found that HLA-G-targeting CAR-NK cells were capable of reducing xenograft tumor growth while extending the median survival in orthotopic mouse models of triple negative breast cancer and glioblastoma. Antitumoral immunity was achieved via restoring native NK cytolytic functions mediated by the anti-HLA-G scFv moiety, which promoted Syk/Zap70 activation in NK cells. The authors also found that pre-treatment with low-dose chemotherapy induced the overexpression of HLA-G, increasing the antitumor efficacy of HLA-G targeting CAR-NK cells both in vitro and in vivo [184].

Another clinical trial is currently evaluating the safety, tolerability, pharmacokinetics, immune response, and preliminary anti-tumor activity of the RO7515629 drug in participants with advanced or metastatic solid HLA-G+ tumors (NCT05769959).

#### 4.1.3. Epigenetic Modulation of MHC I Expression

MHC I expression can be modulated by cancer cells through DNA hypermethylation, histone deacetylation, and the trimethylation of H3K27 on the promoters of heavy chain, B2M, APM components, and NLRC5. Stone et al. found that clinical doses of DNA methyltransferase inhibitors (DNMTi), and histone deacetylase inhibitors (HDACi) on ID8-Defb29/Vegf-a (ID8-VEGF-Defensin) tumor-bearing mice, decreased immunosuppression within the TME by enhancing type I IFN signaling and alleviating PD-1 blockage [185]. DNA hypomethylation by azacytidine, a DNMTi, seems to render EOC tumors more immunogenic and responsive to treatment by inducing the expression of endogenous retroviruses and an innate immune response. This response was characterized using cytosolic sensing of double-stranded RNA causing a type I IFN response and apoptosis, and by modulating chromatin remodeling and gene expression beyond hypomethylation [186,187].

Moufarrij et al. showed that treatment of the ID8 *Trp53^−/−^* EOC murine model with the combination of 5-azacytidine and the histone deacetylase 6 inhibitor Nexturastat A resulted in an amplified type I IFN response, leading to increased cytokine and chemokine expression and a higher level of expression of the MHC I. These changes promote an increased recruitment of IFN-γ+ CD8+, NK, and NKT cells, thereby reversing the immunosuppressive TME and decreasing the tumor burden [188]. Other epigenetic modifiers such as entinostat, an HDACi, combined with azacytidine, significantly increased the expression of MHC II in vitro on ID8 cells, and impeded ovarian tumor growth in MISIIR-TAg mice and in immune competent mice bearing ID8 tumors [189].

The safety and tolerability of CC-486 (an orally bioavailable form of azacytidine) has been assessed in combination with carboplatin and nab-paclitaxel, showing partial responses and stable disease [190]. While CC-486 combined with durvalumab (NCT02811497) did not show robust pharmacodynamics or clinical activity in selected immunologically cold solid tumors [191], other clinical trials are still ongoing/pending results, including the combinations of azacytidine, valproic acid, and carboplatin (NCT00529022), and azacytidine with pembrolizumab (NCT02900560).

### 4.2. Therapeutic Advances Targeting EMT in Cancer

Anti-EMT therapies possess the potential of reducing the invasion and spread of cancer cells, as well as being used in combination with other anti-tumor therapies to enhance patient response [192]. The quest to control the growth of the primary tumor and metastatic dissemination via attenuating EMT has received increasing attention recently [193,194]. Mesenchymal and partial-EMT cancer stem and stem-like cells have been characterized as resistant to drugs and other therapies mediated by increased drug efflux and the downregulation of apoptotic pathways [195]. Mesenchymal cells also upregulate HDACs as a means to stabilize EMT proteins [196,197]; blocking this activity with anti-EMT therapies might, therefore, sensitize mesenchymal cells to chemotherapy [198,199]. An inherent risk of using anti-EMT therapies is the potential to push cells towards the reverse mesenchymal-to-epithelial transition, which may encourage already circulating cancer cells to colonize their immediate surroundings. However, reversion to a more epithelial state could also restore MHC I re-expression resulting in a resurgence of anti-tumoral activity from immune cells. It is also important to remain cognizant that classical EMT-TFs, the putative targets for anti-EMT therapy, induce EMT in a context- and disease-dependent manner, therefore these therapies might not be effective across all cancers [193]. Figure 5 summarizes the EMT effect on the EOC TME.

#### 4.2.1. TGF β1

Several strategies have been proposed to target EMT, with the most progress being made in the direct targeting of EMT-TFs or upstream regulators of the EMT cascade, such as TGF β1. Several anti-TGF β1 therapies are being tested in clinical trials: anti-TGF monoclonal antibody GC1008 in relapsed malignant pleural mesothelioma (NCT01112293), chemotherapy combination with TGF-β blockade drug NIS793 in metastatic pancreatic ductal adenocarcinoma (NCT04935359), hybrid PD-L1/TGF-β bifunctional protein acting as a neutralizing trap for TGF-βRII in advanced stage ovarian cancer (NCT05145569), and radiotherapy in combination with the TGF-β inhibitor LY2157299 in metastatic breast cancer (NCT02538471). The success of these agents remains to be seen; though these drugs are being used to treat highly aggressive and metastatic cancers. Similar therapies that target mutated and dysregulated TGF-β pathway feedback pathways, such as the EGFR pathway are currently participating in clinical trials (NCT04429542, NCT00063401, NCT02954523) or have been brought to market (Cetuximab, Erlotinib) [200,201,202,203]. Other EMT inducers, such as EGF and TNF-α, have also been targeted for therapies with the goal of attenuating EMT and cancer metastasis [193]. While anti-TGF β1 therapies have primarily targeted the reversal of EMT as a way to block metastasis, there may be the added benefit of MHC I re-expression, as described in detail in Section 3.1.

#### 4.2.2. EMT-TFs

Another strategy to attenuate EMT is to target EMT-TFs such as TWIST, ZEB, and SNAIL. Very little research exists in the area of blocking EMT-TFs as a therapy to attenuate metastases. Unfortunately, deeply intertwined feedback networks and the fluctuating expression of various EMT-TFs make them poor druggable targets [204]. Research attention has therefore turned towards epigenetic regulators of EMT-TFs to reversibly attenuate EMT and control the interplay between the dissemination of cancer cells and satellite colony formation. Drugs targeting epigenetic modulators such as HDAC inhibitors and lysine-specific histone demethylase 1 inhibitors are currently in clinical trials, aiming to modify their targets’ ability to bind EMT-TF promoters and thus suppress EMT [205,206].

An alternative strategy to attenuate EMT is to target factors that enable cells with any features of epithelial-mesenchymal plasticity to exert their pro-metastatic actions, such as fibronectin 1 [207,208], vimentin [209], and cadherins [210,211]. In a recent example where pro-metastasis EMT-associated factor fibronectin 1 was targeted, CAR-T therapies targeting fibronectin did not improve the efficacy of the therapy in combination with the induced expression of MHC-I by IFN-γ [207]. This result suggests that further investigation is required into pro-metastasis EMT-associated factors like fibronectin 1 and MHC I expression, where any correlation between them may be incidental.

Overall, a multitude of strategies to block EMT and prevent a primary cancer from metastasizing are being tested in trials targeting many types of cancer. Unfortunately, at this time, very little research has been carried out to examine how these EMT-specific therapies affect the expression of MHC I molecules. Based on the various research studies explored in this review, anti-EMT therapies are expected to have different effects on MHC I machinery depending on the context. Much of the evidence suggests that anti-EMT therapies could be detrimental in ovarian cancer and reduce the bulk expression of classical HLA-A, -B, and -C in the cancer cell population, thus reducing their recognition by the immune system.

## 5. Future Directions and Open Questions

This review focuses on the crucial role of MHC I molecules in ovarian tumor immunity. Despite the fact that MHC I molecules were discovered more than 60 years ago [212,213,214], and that MHC is among the most central fields in basic and clinical immunology [215], there are still many unknowns regarding the modulation of MHC I molecules, both classic and non-classic, in tumor immunity.

It is well established that the selective immune pressure on tumor cells allows for outgrowth of variants with low MHC I expression. However, it is still not clear if, under specific disease state(s), the EMT of the cancer cells could be directly linked to an increase in MHC I expression as shown in Figure 2. It remains unclear if these observations are based on gene expressions which do not necessarily represent protein expression, or whether these results reflect a relationship in human tumors that is not the same as in vitro or mouse model systems.

Since MHC I expression on the cell surface relies on the stability conferred by antigen binding, the more immunogenic the peptide, the longer the lifespan of the MHC I molecule on the cell surface [216,217]. Based on previous findings, we hypothesize that the mesenchymal state could render the cancer cells more immunogenic because the peptidome could be different from a more epithelial cancer cell, especially since more mesenchymal cells are associated with metastasis. However, the extent to which the proteome shapes the peptidome is still largely unknown during tumorigenesis. For instance, protein degradation was validated as an important factor in HLA I presentation [218], but the stability/instability of the proteome of more mesenchymal cancer cells has yet to be studied. Since EOC is known to generate neoantigens poorly, by becoming mesenchymal, cancer cells could perhaps produce more immunogenic peptides that could attract more CTLs to push immunoediting and CTL exhaustion further, as has been recently reported in breast cancer [119]. Therefore, immunopeptidome studies of EOC cells treated with EMT inducers such as TGFβ, compared with their more epithelial counterparts, may help elucidate how the TAAs generated under each context differ, and how they may interact with CTLs.

Numerous enigmas persist regarding the intricate interplay between TMB, its impact on HLA expression, and the subsequent modulation of antitumoral immune responses. The TMB also plays a key role in both stemness and immunogenicity; BRCA1/2 genetic loss in EOC results in differential immunogenicity, with high immune infiltration being associated with PTEN-loss and BRCA1 promoter-methylated tumors, and low immune infiltration with a significantly shortened overall survival [219]. Nevertheless, EOCs with BRCA1/2 mutation and PTEN loss had significantly higher HRD scores, but displayed significantly fewer CD3+, CD8+, and FOXP3+ T cells [219]. In their recent study, Fumet et al. [61] showed that the HLA-E/CD94-NKG2A/2C axis influences antitumoral immunity in HRD ovarian tumors, revealing opportunities to explore and further investigate the interacting immune network in these specific tumor genotypes.

Many open questions remain regarding MHC I expression and tumorigenicity. *How is a more “aggressive” genotype potentially more immunogenic and recognizable by CTLs? Is this a transient state of their stemness reprogramming*? A loss of heterozygosity in HLA I represents a genetic barrier to effective immunotherapy, which would require alternative ways to harness the immune system to maximize clinical benefit [220].

*What is the amplitude of HLA diversity/heterogeneity that would correlate with efficient immune responses in EOC*? F. Garrido previously commented on the different HLA I expression profiles found in tumor clones maintained ex vivo, contrasting them with the homogeneous tumor cell lines used in laboratories, which are often derived from a single clone and therefore lacking HLA “diversity” [2]. Therefore, more relevant models need to be employed to better elucidate the nature of tumor immunogenicity.

*What level of HLA expression influences tumor promotion and EMT? Does the grade and type of EOC affect HLA immune regulation in the TME differently*? Ovarian cancer is not a single disease but comprises various subtypes with different molecular characteristics. The impact of MHC I expression can vary among these subtypes. It is essential to consider the specific subtype and its genetic constitution when evaluating the role of MHC I expression.

Standard treatment centered on neo-adjuvant chemotherapy using carboplatin/paclitaxel in EOC patients showed that HLA I expression in tumors decreases after treatment, hampering T cell recruitment and activation in the TME [221]. Therefore, strategies aiming to upregulate MHC-I expression during or after neoadjuvant chemotherapy could provide better treatment outcomes in these patients [221]. Epigenetic modulators (such as azacytidine) aiming to increase MHC I could be a good approach to increase immunogenicity.

In summary, the relationship between MHC I expression and ovarian cancer is multifaceted and context dependent. Generally, high MHC I expression can be favorable as it may enhance the immune response against the tumor. However, other factors, including the tumor’s genetic constitution, the EMT status of the cancer cells, and the treatment approach, need to be considered when assessing its significance in individual cases. Additionally, the field of cancer immunotherapy is continually evolving, and ongoing research may reveal new insights into the role of MHC I expression in the response of ovarian cancer to treatment.

## Figures and Tables

**Figure 1 cancers-15-05694-f001:**
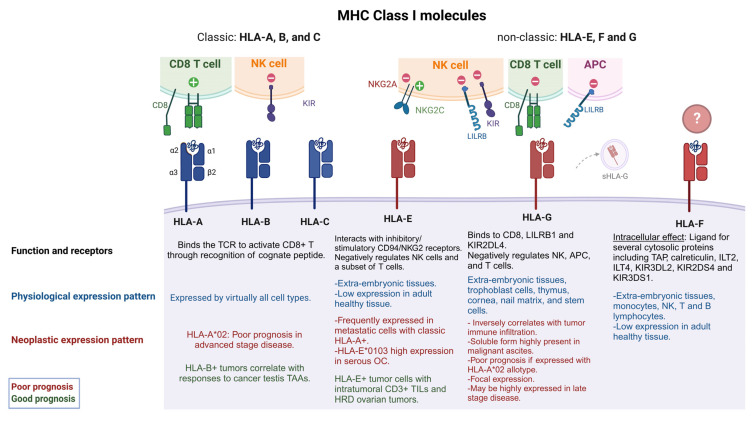
Classic and non-classic HLA I and their role in EOC tumorigenicity. HLA I molecules are formed from three α subunits (α1-3) and one B2M subunit (β2). Classic HLA I molecules include HLA-A, -B and –C while non-classic are known as HLA-E, -F, G. Classic HLA I molecules allow the presentation of endogenous peptides to CD8+ T cells, while non-classic HLA I molecules negatively regulate NK cell function. Some classic HLA Is (HLA-A*02) have been associated with poor EOC prognosis (red captions) when co-expressed with non- classic HLA-G and HLA-E [8]. HLA-E can interact with stimulatory or inhibitory receptors on NK cells and some T cells, while HLA-Gs have a broader immune-modulatory capability, having a negative effect on NKs, T cells and APCs. sHLA-G can also be found in extracellular vesicles and tumor derived exosomes. HLA-E and HLA-G expression levels have been associated with poor prognosis and late stage of disease. However, in some specific cases, HLA-E expression along T cell infiltration and HRD ovarian tumors have shown a good prognosis (green captions). HLA-F may have immune regulatory functions, but its binding partners are still unknown. See text for more details. HLA-A*02 refers to the allele group, TCR (T cell receptor), NK (natural killer cell), antigen presenting cell (APC), OC (ovarian cancer), HRD (homologous recombinant deficiency), TAP (transporter associated with antigen presentation), ILT2 [immunoglobulin (Ig)-like transcript 2], KIR3DS1 (killer cell Ig like receptor three Ig domains and short cytoplasmic tail 1), KIR3DL2 (KIR3D and long cytoplasmic tail 2), KIR2DS4 (KIR two Ig domains and short cytoplasmic tail 4), LILRB1 (Leukocyte Immunoglobulin-Like Receptor B1).

**Figure 2 cancers-15-05694-f002:**
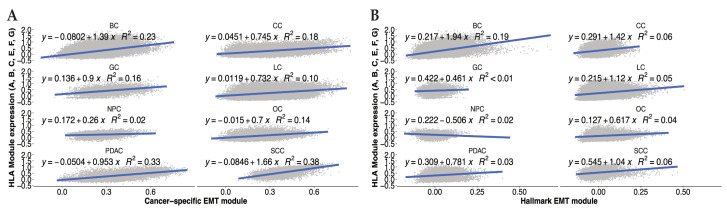
Average gene expression of MHC-I related HLA allotypes correlates with EMT in various cancer types. Expression of MHC-I related HLA allotypes (HLA A-G) scored in the cancer cell compartment of a variety of tumors (BC—Breast Cancer, CC—Colon Cancer, GC—Gastric Cancer, LC—Lung Cancer, NPC—Nasopharyngeal Cancer, OC—Ovarian Cancer, PDAC—Pancreatic Ductal Adenocarcinoma, SCC—Squamous Cell Carcinoma). HLA expression was correlated with EMT scores of those cells based on either (**A**) a cancer-specific EMT module published by Cook and Vanderhyden [120] or (**B**) the Hallmark EMT module from MSigDb.

**Figure 3 cancers-15-05694-f003:**
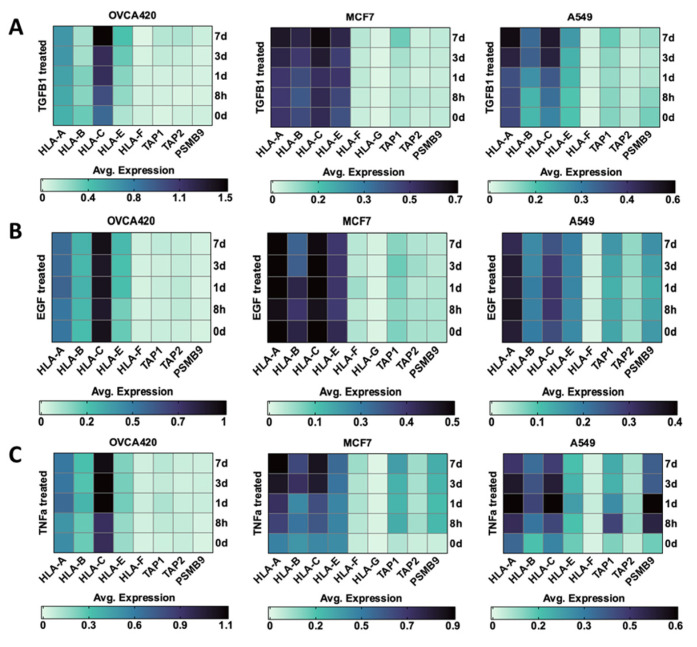
Average gene expression of HLA I allotypes and some APM components correlate with EMT in cancer cell lines. Heat map showing expression levels of transcripts for MHC-I related HLA allotypes (HLA A-G) and machinery (TAP1, TAP2, PSMB9) in ovarian (OVCA420), breast (MCF7), and lung (A549) cancer cell lines in response to treatment with (**A**) TGF β1, (**B**) EGF, (**C**) TNFα for 0 and 8 h, and 1, 3, and 7 days. Data were extracted from single-cell RNA-seq datasets published by Cook and Vanderhyden [130].

**Figure 4 cancers-15-05694-f004:**
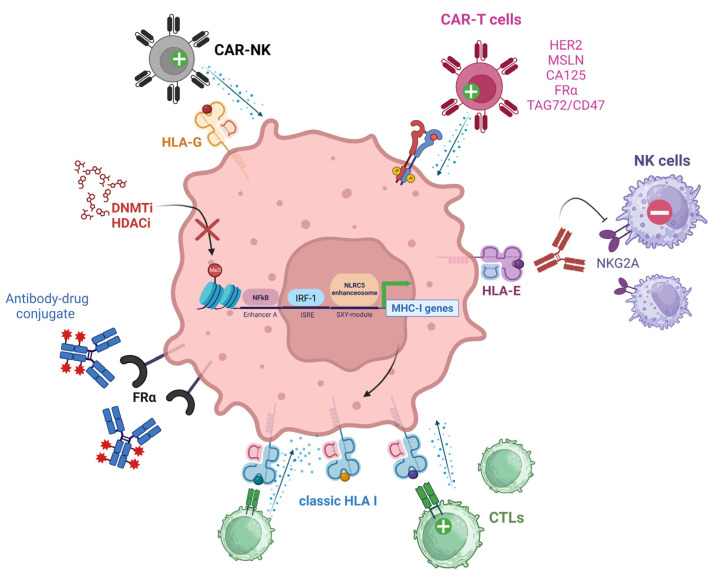
Therapeutic interventions aiming to increase EOC immunogenicity. TAAs highly expressed by EOC cells such as HER2, MSLN, CA125, FRα, and TAG42/CD47 can be targeted with CAR-T cells. FRα can be targeted by direct binding with antibody-drug conjugates. HLA-E+ ovarian tumors can be treated with antibodies blocking the HLA-E/NKG2A axis and impeding NK cell anti-tumoral activity. Poorly immunogenic ovarian tumors can be treated with DNMTi or HDACi agents aiming to decrease DNA hypermethylation or histone deacetylation (indicated by the red “X”). These epigenetic modifiers promote an increase in MHC I gene expression, type I IFN release, and TAA presentation, increasing CTL binding and antitumoral activity. HLA-G+ ovarian tumors can be treated with CAR-NK cells aiming to restore NK cytolytic functions. Green (+) symbols denote activation effect while red (−) symbols denote inhibitory signaling. NK (Natural Killer cell), CAR (Chimeric Antigen Receptor), HER2 (Human Epidermal Growth Factor Receptor 2), MSLN (Mesothelin), CA125 (Cancer Antigen 125), FRα (Alpha-Folate Receptor), TAG72 (Tumor-associated glycoprotein 72), NKG2A (NK group 2 member A receptor), CTLs (Cytotoxic T Lymphocytes), DNMTi (DNA methyltransferase inhibitor), HDACi (histone deacetylase inhibitor), IRF-1 (Interferon Regulatory Factor 1), ISRE (Interferon-Stimulated Response Element), NLRC5 (NLR family CARD domain containing 5).

**Figure 5 cancers-15-05694-f005:**
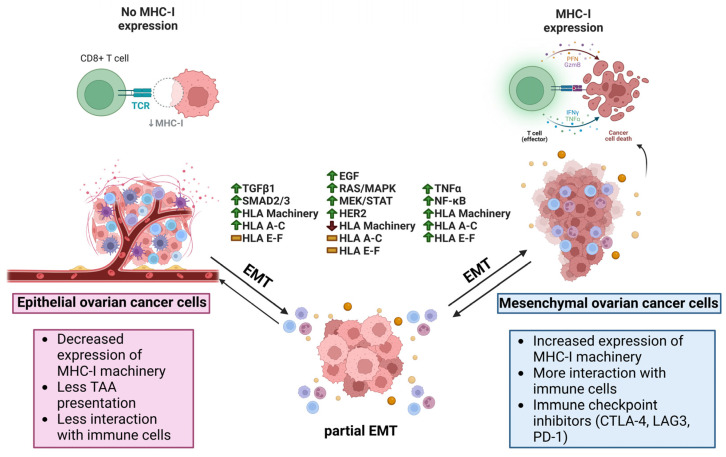
Potential relationship between antigen presentation and EMT in ovarian cancer. In ovarian cancer, epithelial cancer cells have decreased expression of classical HLA I related machinery, less interaction with immune cells, and less TAA presentation. Once stimulated with TGF-β1, EMT can begin increasing the expression of EMT-TFs such as SNAIL, SLUG, ZEB1, and TWIST1/2. At the same time, SMAD2/3 pathways are active, leading to the activation of APM and the re-expression of HLA-A, -B, and -C, and further leading to increased immunogenicity as the cells undergo EMT. If the EGF receptor (EGFR) is stimulated, the cells begin to undergo an EMT coinciding with activation of the RAS/MAPK, MEK/STAT, and HER2 pathways while seeing a reduction in APM and no changes to typical and atypical HLA I allotypes. EMT also beings if cancer cells are stimulated by TNF-α, including an increase in NF-κB that is upstream of APM and both typical and atypical HLA molecules. With increased immunogenicity in the mesenchymal phenotype as a result of more antigen presentation, immune cells like CD8+ T cells are able to effectuate their cytolytic functions on the cancer cells through PFN, GZMB, IFN-γ, and TNF-α. TAA (Tumor associated antigen), TGF-β1 (Transforming growth factor β1), EMT-TF (Epithelial-mesenchymal transition transcription factor), SNAIL (Zinc finger protein SNAI1), SLUG (Zinc finger protein SNAI2), ZEB1 (Zinc finger E-box binding homeobox1), TWIST1/2 (Twist-related protein 1/2), SMAD2/3 (Mothers against decapentaplegic homolog 2/3), APM (Antigen presenting machinery), HLA (Human leukocyte antigen), EGF (Epidermal growth factor), RAS/MAPK (Reticular activating system/Mitogen-activated protein kinase), MEK/STAT (Mitogen-activated protein kinase kinase/Signal transducer and activator of transcription), HER2 (Receptor tyrosine-protein kinase erbB-2), EMT (Epithelial-mesenchymal transition), NF-κB (Nuclear factor kappa-light-chain-enhancer of activated B cells), PFN (Perforin), GZMB (Granzyme B), IFN-γ (Interferon gamma).

## Data Availability

The data presented in this study are available in this article and Appendix A.

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
