# Peer review of "Unveiling the Immunogenicity of Ovarian Tumors as the Crucial Catalyst for Therapeutic Success"

_cancers, 2023, doi:10.3390/cancers15235694_

Round 1

Reviewer 1 Report

Comments and Suggestions for Authors

Title: Unveiling the Immunogenicity of Ovarian Tumors as the Crucial Catalyst for   Therapeutic Success

Authors: GM Rodriguez, E Yakubovich, BC Vanderhyden

Review:  This review focuses on the potential roles of specific components of the immune system in regulating ovarian cancer progression. Specifically, the expression and potential functions of components of the complex immune system in ovarian cancer are reviewed and specific therapeutic approaches related to regulating the complex immune surveillance system are presented.  The list of references and knowledge presented in the review are impressive and a bit overwhelming.  It would be helpful if the authors could provide more detailed illustrations to facilitate this complex and comprehensive overview that is based on complex functions of immune factors and pathways.

For example, although Figure 1 summarizes the components and functions of different parts of the immune system, the complex mechanisms by which these factors act and are regulated are less well presented/imaged.  For example, the receptors for the immune factors and how they function are not always clear. The immune therapeutic approaches (CAR-t) could be demonstrated more clearly with a diagram and some sort of illustration.  Perhaps the future directions could be illustrated by a couple of summary diagrams.

Other comments:

What do the yellow lines in Figure 2 mean?

Models of receptor expression and functions are needed.

Line 531: this is confusing.

Lines 602-602 and elsewhere: what does NCT01004380 and others listed refer to?

The section on the CAR-t approach is interesting but some figures would be helpful to make the underlying mechanisms clearer for an audience that is not totally educated on the immune system.

Line 732: relate to ovarian cancer?

Lines 751-754, lines 758-760,: A model is needed here.

A list of abbreviations would also help to facilitate easier reading and comprehension.

There are some minor grammatical errors.  For example:

Line 11: … is a main factor influencing responses (not response) to cancer therapies

Lines  57-59: expression levels, … along with T cell infiltration … ovarian tumors       have shown

Line 69:  a groove composed of (not by?)

Lines 636-637: this sentence is dense.

Line 669: did not show robust

Line 775: ..represents rather than represent

Comments on the Quality of English Language

Only minor issues as noted in the review

Reviewer 2 Report

Comments and Suggestions for Authors

Rodriguez et al presented a review on Unveiling the Immunogenicity of Ovarian Tumors as the Crucial Catalyst for Therapeutic Success. This study aimed to distinct cancer characteristics related to MHC I expression, immunogenicity, antigen presentation, and epithelial-to-mesenchymal transition. This is the new outlook to correlate the MHC 1 protein. This is the appreciable view by authors to understand the underlying mechanisms. The manuscript can be accepted after the minor corrections of the plagiarised texts.

1. First paragraph on section 2.1 Classic HLA Class I: plagiarism detected. 

2. Page-5: 2nd paragraph needs to be rephrased.

3. An account on Major histocompatibility complex class I-related gene protein (MR1) should be included under non-classical category. 

4. Authors can include 1 table representing the current immunotherapeutic molecules under clinical trials and recent patents .

Round 2

Reviewer 1 Report

Comments and Suggestions for Authors

The authors have nicely responded to all comments and suggestions.